# Recommended distances for physical distancing during COVID-19 pandemics reveal cultural connections between countries

Dongwoo Chai[1], Layla El Mossadeq[1], Michel Raymond[2], Virginie Courtier-Orgogozo[1]*

1 Institut Jacques Monod, Université Paris Cité, CNRS, Paris, France, 2 ISEM, University Montpellier, CNRS, EPHE, IRD, Montpellier, France

* virginie.courtier@normalesup.org

**Data Availability Statement:** Source code and supporting files for analyzing the data are available

## Abstract

During COVID-19 pandemic several public health measures were implemented by diverse countries to reduce the risk of COVID-19, including social distancing. Here we collected the minimal distance recommended by each country for physical distancing at the onset of the pandemic and aimed to examine whether it had an impact on the outbreak dynamics and how this specific value was chosen. Despite an absence of data on SARS-CoV-2 viral transmission at the beginning of the pandemic, we found that most countries recommended physical distancing with a precise minimal distance, between one meter/three feet and two meters/six feet. 45% of the countries advised one meter/three feet and 49% advised a higher minimal distance. The recommended minimal distance did not show a clear correlation with reproduction rate nor with the number of new cases per million, suggesting that the overall COVID-19 dynamics in each country depended on multiple interacting factors. Interestingly, the recommended minimal distance correlated with several cultural parameters: it was higher in countries with larger interpersonal distance between two interacting individuals in non-epidemic conditions, and it correlated with civil law systems, and with currency. This suggests that countries which share common conceptions such as civil law systems and currency unions tend to adopt the same public health measures.

## Introduction

Interpersonal distance, the distance between two interacting individuals, is an essential characteristic of human social interactions [1,2]. The choice of an appropriate interpersonal distance should satisfy two opposite needs, the need to interact with others and the need to maintain a zone of safety to protect the body from potential hazards [2,3]. Interpersonal distance has been shown to vary with several factors including personal relationship with the other person [1], age [4–7], gender [6,8–10], facial expression of the other person [10–12] and country [1,8,13]. A recent study involving 8,943 participants from 42 countries showed that people in warmer countries tend to maintain closer distances toward strangers than in colder countries, but farther toward intimate partners [13].

at a public Github repository: https://github.com/redchai/COVID-Distance-Project.

**Funding:** This research was funded by the European Research Council under the European Community's Seventh Framework Program (FP7/2007-2013 Grant Agreement no. 337579) to VCO. DC was supported by a fellowship from Pohang University of Science and Technology, South Korea (Study Abroad scholarship). The funders had no role in study design, data collection and analysis, decision to publish, or preparation of the manuscript.

**Competing interests:** The authors have declared that no competing interests exist.

**Abbreviations:** CDC, Centers for Disease Control and Prevention; CFA Franc, Franc of the Financial Community of Africa; COVID-19, Coronavirus Disease 2019; GLM, Generalized Linear Model; OxCGRT, Oxford COVID-19 Government Response Tracker; SARS-CoV-2, severe acute respiratory syndrome coronavirus 2; SE, standard error of the mean; WHO, World Health Organization.

Since at least the 1918 influenza pandemic, epidemics have triggered various reactions, including a reduction of interpersonal contacts and an increase in interpersonal distance [14–16]. Avoidance of conspecifics showing signs of disease such as leprosy, polio or fungus infection have been observed not only in humans [17,18] but also in chimpanzees [19], mandrills [20] and frogs [21]. Furthermore, in regions with higher historical prevalence of infectious diseases, people report lower levels of sociosexuality, extraversion and openness [22].

In 1897 bacteriologist Carl Flügge showed that airborne droplets such as those emitted by sick persons while coughing, sneezing or speaking contained infectious germs [23,24]. A three-foot/one-meter recommendation was advocated by WHO and other public health guidelines based on studies done in the 1930s-1950s about the location of droplets and germs after sneezing, coughing and loud talking [25–27]. However, these early studies were limited in their sensitivity. It is only after the SARS outbreak in 2003 and further work documenting more distant spread [28] that certain health agencies, including the Centers for Disease Control and Prevention (CDC) based in the United States, doubled the recommended safe distance from one to two meters.

The severe acute respiratory syndrome coronavirus 2 (SARS-CoV-2) outbreak was declared a global worldwide pandemic by the World Health Organization on 11 March 2020 [29]. The SARS-CoV-2 virus causes coronavirus disease 2019 (COVID-19) and is mainly transmissible via airborne routes through inhalation of aerosols and droplets [30]. When SARS-CoV-2 started to spread across the world, no vaccine nor effective pharmacological interventions were available [31]. Diverse measures were recommended by governments to try to reduce the rate of infection in the general population such as washing hands, wearing face masks, closing schools, lock-downs, praying (e.g. Burundi), daily practicing therapeutic Yoga (e.g. Nepal), reinforcing hygiene for people in contact with camels (e.g. Cameroon) and physical distancing. The Oxford COVID-19 Government Response Tracker (OxCGRT) collected systematic information on various government policy measures including school closures, travel restrictions or vaccination policy [32], with the aim to track and compare policy responses around the world. Overall, a lack of coordination among countries regarding policies and social measures to contain the pandemic was noted by "The Lancet Commission on lessons for the future from the COVID-19 pandemic" in 2022 [31].

Here we examined the minimum distance recommended between people in public spaces (shops, streets, etc.) during the early days of COVID-19 pandemic in the different countries, a measure not analyzed in the OxCGRT tracker. The exact distance and time that SARS-CoV-2 viruses could travel in the air while remaining infectious were not known at the beginning of the pandemic and countries recommended various minimal distances between one meter/three feet and two meters/six feet. Although variation in recommended distances is relatively small, we searched for correlations with parameters describing the outbreak dynamics to test whether choosing different distances had an impact. We also searched for correlations with potentially relevant parameters that might have influenced the decision of health authorities to choose specific recommended distances.

## Materials and methods

### Collection of the recommended distances for each country

The list of all 195 sovereign countries was retrieved from https://unstats.un.org/unsd/methodology/m49/. We also included Hong-Kong and Taiwan in our analysis, even if they are not considered as sovereign countries, because they have their own, independent health agencies. Recommended distances were retrieved individually from the official web pages of each country (S1 Table in S1 File). We performed Google searches in French, English, Spanish and

Arabic using the following keywords: coronavirus, COVID-19, distance, meter, feet and site:.*x* with *x* the top-level internet domain of each country. For other languages, we used Google Translate or asked native researchers from the respective countries for official web pages and recommendations. All recommended distances were collected between 21 May 2020 and 15 June 2020. We managed to collect distances for all countries except Nepal and Tuvalu. Most relevant web pages (S1 Table in S1 File) were also archived at https://web.archive.org/ or https://archive.is/. With respect to recommended distances, our survey is more comprehensive than the COVID-19 Health System Response Monitor (HSRM) tool which collects information about health system pandemic responses across Europe (https://eurohealthobservatory. who.int/monitors/hsrm/hsrm-countries/) [33].

For a few countries, variations were found depending on regions (for example Germany's recommendation was 1.5 m in general but 2 m in some parts of the country) or on settings (for example the 1.5-m-distance observed in Spain could be reduced at open-air events in Catalonia if masks were worn, in Greece the 2m-distance rule could go down to 1.5-m if one wears a mask) (see S1 Table in S1 File for references). For such cases, we used the distance recommended by the government for public spaces in general. When the recommended distance was given both in feet and meters, the one in meters was used for analysis. When the recommended distance was given in imperial units, we changed it into the corresponding metric distance: for countries recommending a separation of 6 feets the value was replaced with 2m, and for 3 feets the value was replaced with 1m. Out of 197 countries, two countries (Nepal and Tuvalu) had no physical distancing recommendation found, nine countries made no precise recommendation in terms of distance and one (South Africa) has all three distances (1, 1.5 and 2m) recommended. Therefore, we used a total of 185 countries with a recommended distance for our analysis.

## Creation of the world map

The recommended distances were visualized on a world map using the rworldmap package in R (https://cran.r-project.org/web/packages/rworldmap/rworldmap.pdf) [34].

## Datasets used for statistical analysis

We examined eleven factors and their relationships with the recommended distances: reproduction rate, number of new cases per million, interpersonal distances (intimate, personal, and social), population density, colonization history, first official language, currency union, legal system, previous exposure to SARS.

For parameters describing the outbreak dynamics during the early stages of the pandemic, two different datasets were used separately. First, effective reproduction rates (Rt) were retrieved from Arroyo-Marioli et al. (2021) [35] (https://github.com/crondonm/TrackingR/ blob/main/Estimates-Database/database.csv), for 131 (May 8th, 2020) or 152 countries (Aug 1st, 2020) depending on the retrieval date. Second, effective reproduction rates and the smoothed numbers of new cases per million were obtained from Ritchie et al. (2020) [36] (https://github.com/owid) for 131 countries (for May 8th, 2020) and 153 countries (for Aug 1st, 2020). Interpersonal distance data (intimate, personal, and social distance) of 42 countries were retrieved from the supplementary file of [13]. Population density for 181 countries was retrieved from file WPP2019_POP_F06_POPULA TION_DENSITY.xlsx uploaded from the United Nations World Population Prospects website https://population.un.org/wpp/ Download/Standard/Population/ on 21 May 2020. Population density of the year 2020 was used for analysis. East Timor was not present in this file. Due to the abnormality of small sovereign nations with excessively high population density, sovereign nations with population

density higher than 1000 people/km$^2$ were excluded from the analysis. For colonization history and official languages of each country, data was retrieved from http://www.cepii.fr/cepii/en/bdd_modele/presentation.asp?id=6. The geo_cepii dataset contains geographical variables for 225 countries in the world, including the languages spoken in the country under different definitions and their colonial links. For colonization history data, we only took into account countries that were colonized by a colonizer country which colonized more than 5 countries in the past to reduce statistical purposes. A total of 133 countries were attributed to one of the following colonizer countries: Great Britain, France, Russia, Turkey, Spain, and Portugal. For the other countries, we attributed the value "other colonizer or not colonized." For official languages, only the first official language used in each country was used for simplicity. Most countries (101 countries) were attributed one of the four most common languages (English, French, Arabic, Spanish) and the other countries were attributed the value "other language".

Table of each legal system was retrieved from http://www.juriglobe.ca/eng/langues/index-alpha.php. Excluding the 59 countries with no recommended distances, non-sovereign countries, one customary country and two muslim countries, a total of 75 civilist countries, 25 common law countries, and 85 mixed countries were used in the statistical analysis.

For currency unions we retrieved data from four currency unions (EURO, CFA Franc, United States Dollar, Eastern Caribbean Dollar) from the following sites: EURO: https://european-union.europa.eu/institutions-law-budget/euro/countres-using-euro_en, CFA FRANC: https://www.imf.org/external/pubs/ft/fabric/backgrnd.html, United States Dollar: https://worldpopulationreview.com/country-rankings/countries-that-use-the-us-dollar, Eastern Caribbean Dollar: https://www.eccb-centralbank.org/p/about-the-eccb. All the other countries that did not have one of these four currency unions in our model were attributed the value "other currency".

The list of the countries which had cases of SARS-CoV infection during the 2002–2003 outbreak was retrieved from: https://www.who.int/publications/m/item/summary-of-probable-sars-cases-with-onset-of-illness-from-1-november-2002-to-31-july-2003.

## Statistical analysis

All statistical analyses were performed using R (version 4.3.1, https://cran.r-project.org/). Logistic regression was used to analyze distance recommendations. For the recommended distance, we used a binary variable, corresponding to either one meter (or three feet) or higher than one meter. Gaussian generalized linear models were used to test whether COVID-19 effective reproduction rate (response variable) or the number of new cases per million (response variable) were affected by the recommended distance for social distancing (explanatory variable) in the early stages of the pandemic in May 2020 and August 2020.

To look for factors that might have influenced the choices of health agencies for a given minimal distance, we used binomial generalized linear models with the recommended distance as the binary response variable. Explanatory variables were interpersonal distances (intimate, personal, and social), population density, colonizer country (Great Britain, France, Russia, Turkey, Spain, Portugal, other), first official language (English, French, Arabic, Spanish, other), currency union (euro, CFA Franc, United States Dollar, Eastern Caribbean Dollar, other), legal system (mixed, common, civil), previous exposure to SARS (yes or no). All quantitative variables were centered. In model 0, we used all these explanatory variables and 40 countries, for which both interpersonal distance data and recommended distance values were available. In model 1, personal and social interpersonal distances were excluded, due to high correlation between the three interpersonal distance values (r > 0.6 for all comparisons). For other cultural factors, first official language and previous colonization history variables showed

high correlation values (cramer's V value >0.25), and thus the language factor was excluded for statistical analysis. Since interpersonal distance only included data from 42 countries, a separate regression model without the intimate interpersonal data was also performed. A total of 40 countries were used in model 1; 175 countries were assessed in model 2. The significance of each explanatory variable was calculated by removing it and comparing the resulting variation in deviance using the $\chi^2$ test, as done by the function Anova from the car R package. The variance inflation factor (VIF) was computed using the Vif function of the R package car.

## Code availability

Source code and supporting files for analyzing the data are available at a public Github repository: https://github.com/redchai/COVID-Distance-Project

## Result

### Most countries recommended minimal distances for physical distancing during COVID-19 pandemic

The official recommendations regarding the distance that people should maintain in public spaces during the COVID-19 pandemic were collected for all countries except Nepal and Tuvalu between 21 May 2020 and 15 June 2020 (S1 Table in S1 File). We found that all governments recommended physical distancing except Afghanistan and Eritrea. Four patterns were identified regarding the minimal recommended distance: no exact values specified (e.g. Sweden), a fixed distance (most countries), a range of values (e.g. Argentina: "between 1 and 2 meters"), diverse values provided by various official institutions (e.g. South Africa: either 1, 1.5 or 2 meters depending on the official institution) (S1 Table in S1 File). Overall, most countries recommended specific distances for physical distancing during COVID-19 pandemic (Fig 1A and 1B). For only 9 countries advising for social distancing we did not find any specified minimum distance: Ecuador, Gambia, Mauritania, Nicaragua, Oman, Sweden, Tonga, United Republic of Tanzania and Uruguay.

In a few countries, the recommended distance was officially changed to a lower value as the contamination risk was deemed to decrease. For example, Denmark switched from a 2-m rule to a 1-m rule in May 2020 [37] and Switzerland switched from 2 to 1.5m in June 2020 [38]. Overall, the most common guideline, observed in 45% of the countries, was 1m or 3 feet, which was also the World Health Organization recommendation to "keep physical distance of at least 1 meter from others, even if they don't appear to be sick" [39]. The highest recommended distance was 2m or 6 feet, prescribed in 31% of the countries (Fig 1A and 1B).

### The recommended distance had no clear effect on COVID-19 outbreak dynamics

To test whether countries recommending higher distances had lower transmission rates, we performed binomial GLM using as the response variable either the effective reproduction number or the smoothened number of new COVID-19 cases per million (Figs 2 and 3). With data from May 2020, a higher recommended distance was associated with a lower transmission rate, as expected, although this effect was marginally non-significant (Table 1, $\beta$ = -0.912, SE = 0.48, P = 0.052, model using Arroyo-Marioli et al. data [35], $\beta$ = -0.88, SE = 0.48, P = 0.06, model using Ritchie et al. data [36]). Furthermore, no effect was detected on the smoothened number of new cases per million ($\beta$ = 0.001, SE = 0.004, P = 0.784, model using Ritchie et al. Data [36]). Data from Aug 2020 depicted the opposite pattern, with a higher recommended distance significantly associated with a higher reproduction rate ($\beta$ = 1.50, SE = 0.54,

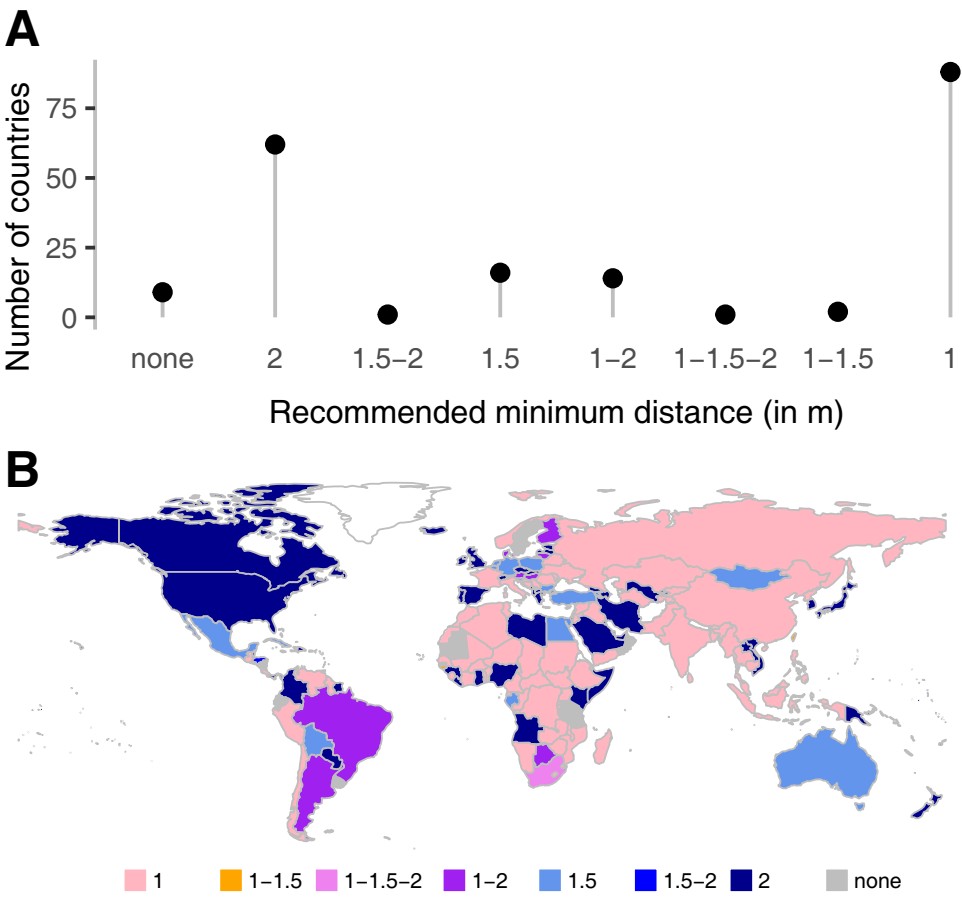

**Fig 1. Distribution of the recommended distances for each country.** (A) The y-axis indicates the total number of countries for each recommended minimal distance for physical distancing. The 3- and 6-feet distances were converted into 1- and 2-m distances, respectively. There are only two countries which recommended 1–1.5 m (Guinea-Bissau and Taiwan), one 1, 1.5 and 2 m (South Africa) and one 1.5–2 m (Honduras). (B) Worldwide map of the recommended minimal distances for each country.

P = 0.004, model using Arroyo-Marioli et al. data [35], $\beta$ = 1.52, SE = 0.54, P = 0.003, model using Ritchie et al. data [36]) and a higher number of new cases per million, although with a low coefficient value ($\beta$ = 0.01, SE = 0.004, P = 0.007, model using Ritchie et al. Data [36]).

## Recommended distance correlates with interpersonal distance, legal system and currency union

A world map view of the recommended distances (Fig 1B) did not reveal any obvious geographical pattern, whether related to latitude, longitude or distance to the outbreak epicenter in China. We hypothesized that the choice made by diverse health agencies across the world for a specific recommended distance might be influenced by different factors. First, recommended distances may be higher in cultures where people tend to be more distant from each other in general. Second, several newspaper and blog articles reported that it was difficult to respect the one-meter distance in highly populated markets and public transportation in Madagascar and Burundi [40–42] so we wondered whether countries with higher population density would tend to recommend lower distances. Third, we noted that Faroe islands changed their recommendation from 2m to 1m [43] three days after their former colonizer country

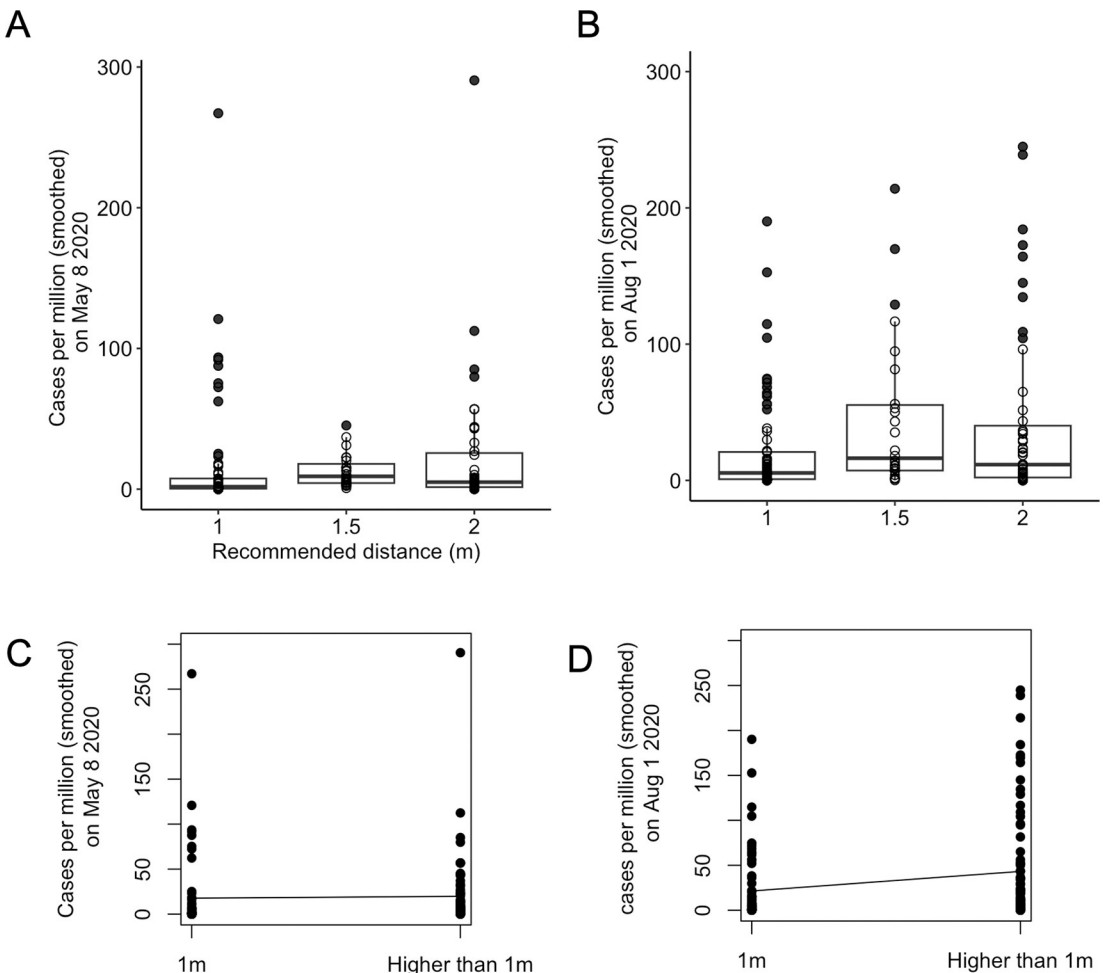

**Fig 2. Boxplot & Generalized linear model of the number of new cases per million.** Boxplot of the smoothened number of new cases per million in May 8th, 2020 (A) and in Aug 1st, 2020 (B) plotted against the recommended distance during COVID-19 pandemic. Gaussian generalized model of the smoothened number of new cases per million for May 8th, 2020 (C) and for Aug 1st, 2020 (D).

Denmark did [37], so we speculated that former colonized countries may usually have continuing links with their colonizer regarding healthcare management and may tend to use the same recommended distances. Noting that both the United States of America and the United Kingdom recommended 6 feet, we hypothesized that former British colonies or English-speaking countries may also tend to recommend such a distance. Along with this, given that the common law system emerged in medieval England and spread among British colonies across the globe [44], we tested whether the recommended distances varied according to the three main legal systems (civil, common, mixed). Finally, we hypothesized that countries who suffered SARS-CoV-1 cases in 2003 would possibly recommend a higher distance. We thus decided to test the influence of the following variables: interpersonal distances (intimate, personal, and social), population density, colonizer country (Great Britain, France, Russia, Turkey, Spain, Portugal, other), first official language (English, French, Arabic, Spanish, other), currency union (euro, CFA Franc, United States Dollar, Eastern Caribbean Dollar, other), legal system (mixed, common, civil), previous exposure to SARS (yes or no).

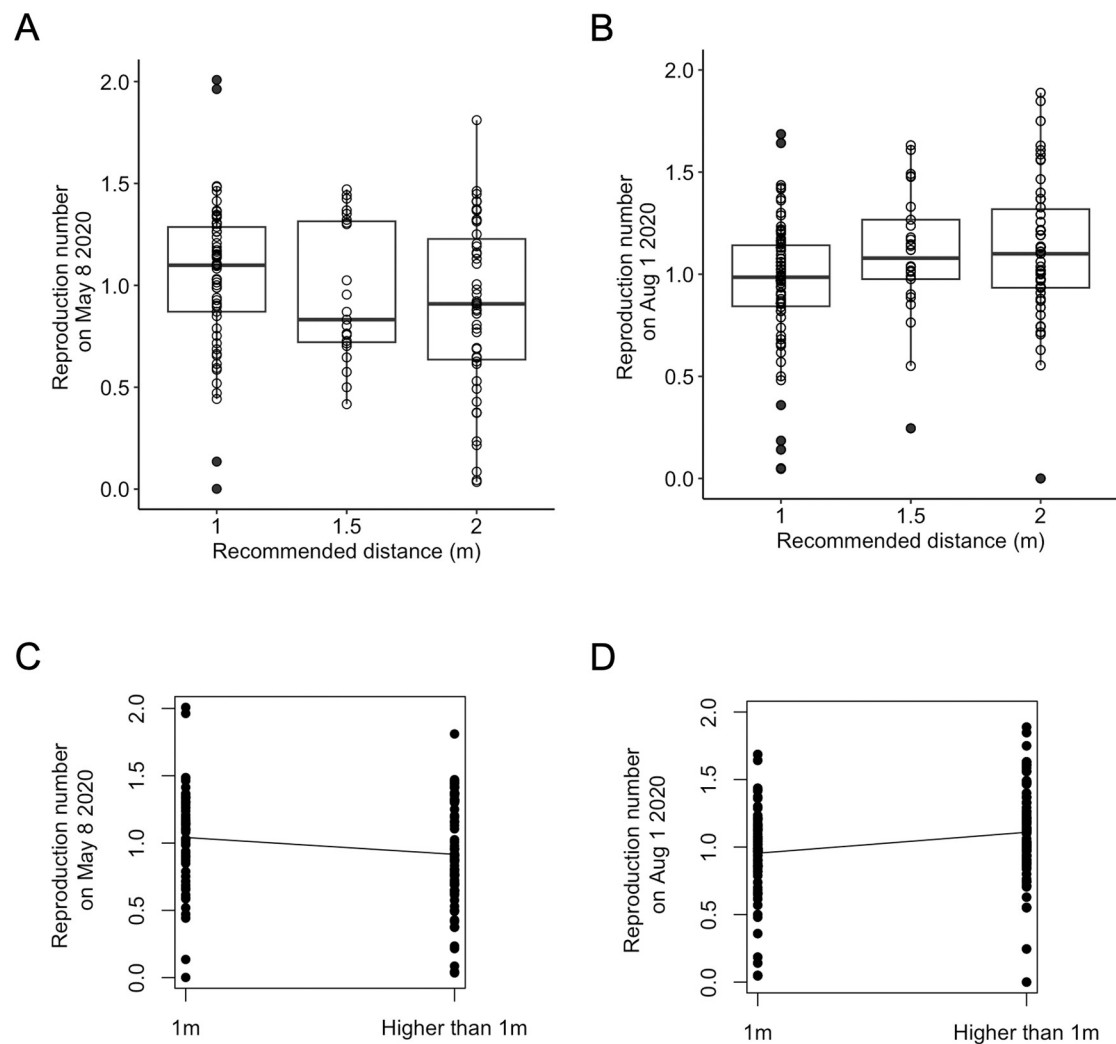

**Fig 3. Boxplot & Generalized linear model of the effective reproduction number $R_t$ (model using Arroyo-Marioli et al. data).**
Boxplot of the estimated effective reproduction rate in May 8th, 2020 (A) and in Aug 1st, 2020 (B) plotted against the recommended distance during COVID-19 pandemic. Gaussian generalized linear model of the estimated effective reproduction rate in May 8th, 2020 (C) and in Aug 1st, 2020 (D) according to the recommended minimal distance.

To test whether the probability that a country had a recommended minimum distance of 1 m, or higher than 1 m, is influenced by these variables, binomial generalized linear models (GLM) were made. Model 0 considered the 42 countries for which three interpersonal distances (intimate, personal, and social) have been documented [13]. The three interpersonal distance variables were highly correlated with each other (Pearson's r > 0.6 for all comparisons). We found that the intimate distance marginally correlated with the recommended distance (one-way ANOVA test, P = 0.074) whereas the social and personal interpersonal distances, i.e. the average face-to-face distances with a stranger and an acquaintance, respectively, did not correlate significantly with the recommended distance (one-way ANOVA test, social distance: P = 0.851, personal distance: P = 0.407). In addition, the languages variable correlated with the currency union variable (Cramer's V value >0.25). We thus removed the personal and social interpersonal distances as well as the languages variable from Model 0 and obtained Model 1. With this model, the intimate distance showed a significant effect on

**Table 1. Effect of recommended distance on the dynamics of COVID-19 pandemic.** The estimated standard error of the mean (SE), $\chi^2$ statistics, degree of freedom (df), and corresponding p-values are given. Bold characters indicate significant (P<0.05) effects.

| | 8 May 2020 | | | | 1 August 2020 | | | |
|---|---|---|---|---|---|---|---|---|
| | Model using Arroyo-Marioli et al. data | | | | Model using Arroyo-Marioli et al. data | | | |
| | β | SE | $\chi^2$ (df) | $P\ (>\chi^2)$ | β | SE | $\chi^2$ (df) | $P\ (>\chi^2)$ |
| Intercept | 1.05 | 0.05 | | | 0.95 | 0.04 | | |
| Binomial recommended distance | -0.13 | 0.07 | 3.78(1) | 0.052 | 0.16 | 0.05 | 8.50(1) | **0.004** |
| | Model using Ritchie et al. data | | | | Model using Ritchie et al. data | | | |
| Intercept | 1.04 | 0.05 | | | 0.95 | 0.04 | | |
| Binomial recommended distance | -0.13 | 0.07 | 3.53(1) | 0.06 | 0.16 | 0.05 | 8.87(1) | **0.003** |
| | Number of new cases per million (smoothened) | | | | Number of new cases per million (smoothened) | | | |
| Intercept | 17.79 | 5.27 | | | 21.40 | 5.79 | | |
| Binomial recommended distance | 1.95 | 7.21 | 0.07(1) | 0.786 | 21.77 | 8.17 | 7.11(1) | **0.008** |

recommended distance ($X^2$ = 3.96, df = 1, P = 0.047, Table 2 and S2 Fig) in the expected direction, with recommended distance increasing with higher intimate distance. Also, currency union showed a significant effect on recommended distance ($X^2$ = 6.10, df = 2, P = 0.047,

**Table 2. Effects of different variables on the probability of having a recommended distance of one versus more than one meter for two different models.** The estimate (β), standard error of the mean (SE), $\chi^2$ statistics (degree of freedom (df)), and corresponding p-values are given. For each qualitative variable, the modality included as the intercept is indicated in parentheses in the first column. Bold characters indicate significant (P < 0.05) effects.

| | Model 1 with 42 countries, including intimate interpersonal distance | | | | Model 2 with 175 countries, without interpersonal distance | | | |
|---|---|---|---|---|---|---|---|---|
| | β | SE | $\chi^2$ (df) | $P(>\chi^2)$ | β | SE | $\chi^2$ (df) | $P(>\chi^2)$ |
| Intercept | -3.37 | 3.39 | | | -0.68 | 0.50 | | |
| Population density | <0.001 | <0.01 | 0.19(1) | 0.67 | <0.001 | <0.01 | 1.17 (1) | 0.28 |
| Colonization (Other colonies) | | | 5.83(5) | 0.32 | | | 4.80 (6) | 0.57 |
| Spain | 0.61 | 1.60 | | | 0.07 | 0.68 | | |
| France | - | - | | | -0.05 | 0.66 | | |
| United Kingdom | -1.61 | 1.81 | | | 0.52 | 0.58 | | |
| Portugal | 17.71 | >10 | | | 0.11 | 0.91 | | |
| Russia | -3.28 | 2.98 | | | -0.71 | 0.74 | | |
| Turkey | -3.28 | >10 | | | 1.20 | 0.92 | | |
| Legal system (Mixed law) | | | 4.34(2) | 0.11 | | | 13.64(2) | **0.001** |
| Civil | -6.38 | 1.63 | | | 0.85 | 0.52 | | |
| Common Law | 19.6 | >10 | | | 2.44 | 0.87 | | |
| Currency union (Other currency) | | | 6.10(2) | **0.047** | | | 13.09(4) | **0.01** |
| CFA franc | - | - | | | -1.30 | 0.88 | | |
| USD | 2.36 | >10 | | | 0.47 | 1.03 | | |
| ECD | - | - | | | 0.08 | 1.35 | | |
| Euro | 3.56 | 1.75 | | | 2.20 | 0.82 | | |
| Exposure to SARS-CoV-1 (No) | | | 5.22(1) | **0.022** | | | 0.07(1) | 0.79 |
| Yes | -2.74 | 1.43 | | | -0.15 | 0.55 | | |
| Intimate distance | 0.09 | 0.05 | 3.96(1) | **0.047** | - | - | - | - |

Table 2): countries with euro and United States Dollar as their official currency were likely to recommend a higher distance. In addition, countries with previous exposure to SARS-CoV-1 tended to show a lower recommended distance for COVID-19 ($X^2$ = 5.22, df = 1, P = 0.022, Table 2), which is opposite to what one would have expected. Other variables had no influence on the recommended distance (P > 0.05, Table 2). All VIF (variance inflation factors) values were below 5 (max observed = 1.51), suggesting an absence of substantial correlation among the explanatory variables.

Model 2 considered 175 countries, but without the interpersonal distance variable. A slight but non-significant effect of population density was observed, in a direction opposite to our expectation, with larger population densities associated with higher recommended minimal distances (Table 2 and S3A and S3B Fig). The average population density for countries which recommended 1, 1.5 and 2 meters were 133.9, 113.5 and 158.7 inhabitants per square kilometers, respectively, and the three groups showed no statistical significance in a one-way ANOVA test (P = 0.377, S3A Fig). The legal system variable showed a significant effect on the recommended distance ($X^2$ = 13.64, df = 2, P = 0.001, Table 2), with countries with a common law system likely to recommend a higher distance, and countries with a civil law or mixed law system likely to recommend a lower distance (S4A Fig). The currency factor showed a significant effect on recommended distance ($X^2$ = 13.09, df = 4, P = 0.01, Table 2). Countries with CFA Franc and the Eastern Caribbean Dollar as their official currency tended to recommend a lower minimum distance, whereas countries with United States Dollar or Euro as their official currency were likely to recommend a higher minimum distance (S4D Fig). Other variables had no influence on the recommended distance (P > 0.05, Table 2 and S4B and S4C Fig). All VIF values were below 5 (max observed = 1.28), suggesting an absence of substantial correlation among the explanatory variables.

## Discussion

In this study we collected the minimum distances recommended by the diverse countries for public spaces during the early days of COVID-19 pandemic. Although the exact distance at which a person could be contaminated by another person was not known at the time and was manifestly expected to vary between settings, we found that for most (94%) of the countries the minimal distance recommendation was a precise number, between 1 meter/3 feet and 2 meters/3 feet. We can think of at least two reasons why health agencies may have chosen to provide a precise figure in their social distancing advice. First, a specific number can help to communicate information in a clear and straightforward manner, facilitating public understanding and avoiding confusion. Second, a precise number can create a sense of accuracy, command or authority. With an exact figure, the information may appear more reliable and credible to the public. Overall, people might be more willing to follow precise guidance rules with figures because they are easier to understand and seem more reliable. In Germany and Switzerland, despite the separation into several counties/cantons which have their own local health agencies, efforts were made so that a single minimal distance was recommended for all citizens at the national level [45,46]. We note however that in certain countries advising a precise minimal distance such as 1 meter or 2 meters the physical distancing recommendations were not always consistent. For example, in South Africa the distance value varied between health agencies (S1 Table in S1 File). In any case, the fact that such precise numbers were provided in most of the countries facilitated our analysis of trends and our assessment of the potential impact of these social distancing measures.

During COVID-19 pandemic, there have been numerous disagreements within the scientific community regarding the sizes of the SARS-CoV-2 containing particles (aerosols,

droplets), the distance they traveled and the time they stayed in the air [47]. Once physical distancing measures were taken by most countries, advanced epidemiological analyses revealed that in certain settings such as a bus [48] or a restaurant [49] SARS-CoV-2 virus could contaminate an individual located at more than 2 meters from an infected person. Yet after these findings were published, no country chose to increase their recommended minimal distance to values higher than 2 meters. This shows that health guidelines take into account not only scientific data but also practical aspects and a certain level of disturbance.

Multiple cross-country analyses have been done to determine the effectiveness of various non-pharmaceutical interventions against COVID-19. For example, a comparison of 202 countries around the world indicated that lockdowns were effective in reducing the number of COVID-19 infections at 10 days and at 20 days after implementation [50]. Other studies found that banning the gathering of <10 people and restricting internal movement were both associated with reduced SARS-CoV-2 transmission [51,52]. To our knowledge, our study is the first one to assess the effectiveness of the minimal physical distance recommendation adopted by various countries on COVID-19 pandemic. Most previous analyses on the potential effects of physical distancing were country-specific and examined differences between participants, irrespective of the government policies. They found either no effect or a reduction in transmission rate and in COVID-19 related mortality [53–56]. Another study across 41 countries found that the average preferred interpersonal distance with a stranger, collected by surveying the population, negatively correlated with the spread of SARS-CoV-2 (measured as the total number of SARS-CoV-2 cases 20 days after the 100th case) [57]. Our results are not as clear. We found that higher recommended distances were associated with either a small, barely significant reduction in COVID-19 dynamics, as expected, or even an increased COVID-19 dynamics, depending on the day of the pandemic we examined. Due to confounding and measurement bias, this analysis should be interpreted with caution. Other policies, including lockdowns, wearing masks, or the compliance of the general public to governmental policies may vary according to cultural factors. It is thus possible that the slight impact of physical distancing that we detected in the present study at the beginning of the pandemic is actually due to other public health measures that correlated with physical distancing measures. An outbreak dynamics in a country is undoubtedly the result of multiple interacting factors, including health policies, individual behaviors, percentage of infectious people coming from other countries, virus intrinsic parameters, etc. An alternative method to assess the effectiveness of the physical distancing measures would be to compare reproduction numbers before and after the minimal distance was officially reduced in certain countries such as Denmark and the United Kingdom. However, this policy change was generally accompanied by the relaxation of a series of other measures such as restaurants and hotels reopening [37,58], and was concomitant with the appearance of new, more transmissible SARS-CoV-2 strains [59]. So these confounding factors make it difficult to estimate the exact effect of a reduction in the official minimal recommended distance.

Classifying countries into two groups, those recommending one meter and those recommending a distance higher than one meter, we tried to identify factors that might have contributed to the choice of one distance or another. Although interpersonal distance data was only available for 42 countries, we found that intimate interpersonal distance displayed a positive correlation with recommended distance, whereas the social and personal interpersonal distances did not correlate significantly with the recommended distance. Retrospectively, it may make sense that the official recommendations tend to follow the intimate distances, rather than distances to strangers or acquaintances, because most of the COVID-19 infections were due to families and close contacts. For example, among 344 clusters involving 1308 cases during the first weeks of the COVID-19 pandemic in China, most clusters (78%-85%) were found

in families [60]. Two possible explanations can account for the correlation between recommended official distances and interpersonal distances. First, health authorities may be influenced by the actual distances observed between fellow citizens and may recommend distances that are more likely to be followed by the general public. Second, upstream undetermined factors that may vary between countries may influence both the interpersonal distance and the distance recommended by health agencies. For example, it has been hypothesized—though not confirmed—that in warmer climates people tend to maintain closer distances toward strangers than in colder countries [13]. It would be interesting to gather interpersonal distances for all countries and to test whether the correlation we detected with the recommended distance during COVID-19 pandemic still holds. Noticeably, it is possible that the physical distancing measures during the COVID-19 epidemic, which have been worldwide and lasted several months, have influenced later habits, so that people who lived in countries advising longer distances during COVID-19 pandemic will tend to exhibit higher interpersonal distances in the future than people who were in countries recommending smaller distances.

We found that legal systems had a major effect in predicting the countries' recommended distance, even after controlling for confounding effects such as population density, previous colonizer country, language, and previous exposure to SARS-CoV-1. The legal system was originally divided into two families, common law, and civil law [44]. Common law emerged in England and spread among British colonies across the globe. Civil law was developed in continental Europe during the Middle Ages and was applied in the colonies of European imperial powers. A third family of mixed legal systems was defined later to improve the classification of legal systems [61]. The general influence of certain legal systems on having similar administrative policies is poorly studied but still generally assumed, due to the legal system being a result of the combination of history, culture, and politics. We can hypothesize that countries might have made their distancing policy based on the data of several other countries which have previously issued their policy, where their legal systems were mostly the same. A notable example would be the issuance of a distancing measure of 1 meter by Senegal on March 2, 2020, and the following same policy of ten other African countries which used the same civil law system and the same currency (CFA Franc) in the following months, with the exception of Guinea-Bissau (1.25m) and Gabon (1.5m) (S1 Table in S1 File). Whether such potential links between countries can also be identified when examining a larger range of health measures such as school closures, travel bans, etc. remains to be done. Overall, our analysis of physical distancing recommendations unravels connections between countries that may lead them to propose comparable health measures.

## Conclusions

Most countries recommended a precise minimal distance for physical distancing during COVID-19 pandemic, with 45% advising one meter/three feet and 49% a higher minimal distance. Our mixed binomial generalized linear models reveal that the average interpersonal distance between two interacting individuals in non-epidemic conditions in a given country correlates with the recommended minimal distance. Furthermore, countries sharing certain cultural factors such as legal system and currency tended to adopt the same distancing values. Studies such as ours can help to understand how decisions are taken at the national level and how they can be influenced by existing connections with other countries.

## Supporting information

**S1 Fig. Boxplot & Generalized linear model of the recommended distance and the effective reproduction number $R_t$ (model using Ritchie et al. data).** Boxplot of the estimated effective

reproduction rate in May 8th, 2020 (A) and in Aug 1st, 2020 (B) plotted against the recommended distance during COVID-19 pandemic. Gaussian generalized linear model of the estimated effective reproduction rate in May 8th, 2020 (C) and in Aug 1st, 2020 (D) according to the recommended minimal distance.
(PDF)

**S2 Fig. Interpersonal distance of 42 countries plotted against the recommended distance during COVID-19 pandemic.** Each point represents one country. Countries with a recommended distance of 1–2 m were considered as 1.5m in this graph. The y-axis indicates the social (A), personal (B) and intimate (C) interpersonal distances reported in [13].
(PDF)

**S3 Fig. Recommended distances and population density.** (A) Boxplot of population density according to the recommended distance. Each point represents one country. Countries with a recommended distance of 1–2 m were considered as 1.5m in this graph. (B) The y-axis indicates the probability to be at the highest recommended distance (higher than 1m) from the binomial generalized linear model. Points represent data for 174 countries.
(PDF)

**S4 Fig. Recommended distances with respect to cultural parameters.** (A) Average recommended distance for civil law (n = 75), common law (n = 21), mixed (n = 85) countries. (B) Average recommended distance for countries previously colonized by Spain (n = 17), France (n = 26), Great Britain (n = 60), Portugal (n = 7), Russia (n = 14) and Turkey (n = 9). (C) Average recommended distance for countries previously exposed (n = 27) or not (n = 157) to SARS-CoV-1. (D) Average recommended distance for countries using CFA Franc (n = 14), Eastern Caribbean Dollar (n = 6), Euro (n = 19), United States Dollar (n = 8). Bars indicate the standard errors. **: $p < 0.05$, ***: $p < 0.01$, 1-way ANOVA followed by Tukey's multiple comparisons test.
(PDF)

**S1 File. Raw data and R script.**
(ZIP)

## Acknowledgments

We thank Julio Bendezu-Sarmiento, Arzu Celik, Horacio Frydman, Marianthi Karageorgi, Rezaul Karim, Bruno Lemaitre, Manon Monier, Antónia Monteiro, Olga Nagy, Magnus Nordborg, Francesca Pinton, Laras Pitayu-Nugroho, Rosina Savisaar and Yumiko Suto for helping us to retrieve recommended distances for certain countries. We are grateful to Aurélie Surtel for archiving web pages at https://web.archive.org/. We thank Olivier Tenaillon for his help with COVID-19 transmission rate data.

## Author Contributions

**Conceptualization:** Virginie Courtier-Orgogozo.

**Data curation:** Dongwoo Chai, Layla El Mossadeq, Michel Raymond, Virginie Courtier-Orgogozo.

**Formal analysis:** Dongwoo Chai, Michel Raymond, Virginie Courtier-Orgogozo.

**Funding acquisition:** Virginie Courtier-Orgogozo.

**Investigation:** Dongwoo Chai.

**Methodology:** Dongwoo Chai, Michel Raymond, Virginie Courtier-Orgogozo.

**Supervision:** Virginie Courtier-Orgogozo.

**Visualization:** Dongwoo Chai, Virginie Courtier-Orgogozo.

**Writing – original draft:** Dongwoo Chai, Virginie Courtier-Orgogozo.

**Writing – review & editing:** Dongwoo Chai, Michel Raymond, Virginie Courtier-Orgogozo.

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
