## [Decision Letter · Decision Letter 0]

3 Oct 2023

PONE-D-23-23810Recommended distances for physical distancing during COVID-19 pandemics reveal cultural connections between countriesPLOS ONE

Dear Dr. Orgogozo,

Thank you for submitting your manuscript to PLOS ONE. After careful consideration, we feel that it has merit but does not fully meet PLOS ONE’s publication criteria as it currently stands. Therefore, we invite you to submit a revised version of the manuscript that addresses the points raised during the review process. In particular, I would like you to clarify if the original data are available, and in case where or why not, and if there was there an association between recommended distance during COVID and mean traditional social distance face to face with strangers. Moreover, please consider expanding the cited literature review. While mostly common knowledge, please specify that social distancing and lockdowns are effective in reducing COVID-19 cases (Alfano, V., Ercolano, S. 2020. The Efficacy of Lockdown Against COVID-19: A Cross-Country Panel Analysis. Appl Health Econ Health Policy 18: 509–517) and the role played by social capital in affecting COVID-19 spread (that is likely to affect mean social distance). The subject is discussed in Alfano, V. (2022). Does social capital enforce social distancing? The role of bridging and bonding social capital in the evolution of the pandemic. Econ Polit 39: 839–859.

We look forward to receiving your revised manuscript.

Kind regards,

Vincenzo Alfano

Academic Editor

PLOS ONE

5. We notice that your supplementary figures are included in the manuscript file. Please remove them and upload them with the file type 'Supporting Information'. Please ensure that each Supporting Information file has a legend listed in the manuscript after the references list.

Reviewers' comments:

Reviewer's Responses to Questions

**Comments to the Author**

1. Is the manuscript technically sound, and do the data support the conclusions?

Reviewer #1: Yes

2. Has the statistical analysis been performed appropriately and rigorously? 

Reviewer #1: Yes

3. Have the authors made all data underlying the findings in their manuscript fully available?

Reviewer #1: No

4. Is the manuscript presented in an intelligible fashion and written in standard English?

Reviewer #1: Yes

5. Review Comments to the Author

Reviewer #1: Great article!

Are the original data available?

Was there an association between recommended distance during COVID and mean traditional social distance vis a vis strangers in the countries where these data were available?

6. PLOS authors have the option to publish the peer review history of their article (what does this mean?). If published, this will include your full peer review and any attached files.

Reviewer #1: **Yes: **Mary V Seeman MD

---

## [Author Response · Author response to Decision Letter 0]

17 Nov 2023

Dear Editor,

Please find our point-by-point response to the reviewers' comments in our cover letter.

With best regards,

Virginie Courtier, on behalf of all the authors

---

## [Decision Letter · Decision Letter 1]

22 Nov 2023

Recommended distances for physical distancing during COVID-19 pandemics reveal cultural connections between countries

PONE-D-23-23810R1

Dear Dr. Orgogozo,

We’re pleased to inform you that your manuscript has been judged scientifically suitable for publication and will be formally accepted for publication once it meets all outstanding technical requirements.

Kind regards,

Vincenzo Alfano

Academic Editor

PLOS ONE

Additional Editor Comments (optional):

Reviewers' comments:

Reviewer's Responses to Questions

**Comments to the Author**

1. If the authors have adequately addressed your comments raised in a previous round of review and you feel that this manuscript is now acceptable for publication, you may indicate that here to bypass the “Comments to the Author” section, enter your conflict of interest statement in the “Confidential to Editor” section, and submit your "Accept" recommendation.

Reviewer #1: All comments have been addressed

2. Is the manuscript technically sound, and do the data support the conclusions?

Reviewer #1: Yes

3. Has the statistical analysis been performed appropriately and rigorously? 

Reviewer #1: Yes

4. Have the authors made all data underlying the findings in their manuscript fully available?

Reviewer #1: Yes

5. Is the manuscript presented in an intelligible fashion and written in standard English?

Reviewer #1: Yes

6. Review Comments to the Author

Reviewer #1: Not required since I answered "all comments have been addressed" above.

Not required since I answered "all comments have been addressed" above.

Not required since I answered "all comments have been addressed" above.

Not required since I answered "all comments have been addressed" above.

Not required since I answered "all comments have been addressed" above.

Not required since I answered "all comments have been addressed" above.

Not required since I answered "all comments have been addressed" above.

Not required since I answered "all comments have been addressed" above.

Not required since I answered "all comments have been addressed" above.

Not required since I answered "all comments have been addressed" above.

Not required since I answered "all comments have been addressed" above.

7. PLOS authors have the option to publish the peer review history of their article (what does this mean?). If published, this will include your full peer review and any attached files.

Reviewer #1: **Yes: **Mary V Seeman

---

## [Editor Report · Acceptance letter]

7 Dec 2023

PONE-D-23-23810R1 

Recommended distances for physical distancing during COVID-19 pandemics reveal cultural connections between countries 

Dear Dr. Courtier-Orgogozo:

I'm pleased to inform you that your manuscript has been deemed suitable for publication in PLOS ONE. Congratulations! Your manuscript is now with our production department. 

Kind regards, 

on behalf of

Dr. Vincenzo Alfano 

Academic Editor

PLOS ONE